# The Effects of Post Heat Treatment on the Microstructural and Mechanical Properties of an Additive-Manufactured Porous Titanium Alloy

**DOI:** 10.3390/ma13030593

**Published:** 2020-01-28

**Authors:** Guisheng Yu, Zhibin Li, Youlu Hua, Hui Liu, Xueyang Zhao, Wei Li, Xiaojian Wang

**Affiliations:** 1Institute of Advanced Wear & Corrosion Resistant and Functional Materials, Jinan University, Guangzhou 510632, China; ygs@stu2017.jnu.edu.cn (G.Y.); jnu-ethan@foxmail.com (Z.L.); huayoululaoda@163.com (Y.H.); liuhuijnu@163.com (H.L.); Hc_sunboy@163.com (X.Z.); liweijnu@126.com (W.L.); 2National Joint Engineering Center of High-performance Wear-resistant Metallic Materials, Jinan University, Guangzhou 510632, China

**Keywords:** heat treatment, Ti-6Al-4V, porous structure, microstructure, mechanical properties

## Abstract

In this work, Ti-6Al-4V (Ti64) porous structures were prepared by selective laser melting (SLM), and the effects of post heat treatment on its microstructural and mechanical properties were investigated. The results showed that as SLM samples were mainly composed of needle-like α′ martensite. Heat treatment at 750 °C caused α′ phase to decompose, forming a lamellar α+β mixed microstructure. As the heat treatment temperature increased to 950 °C, the width of lamellar α phase gradually increased to 3.1 μm. Heat treatment also reduced the compressive strength of the samples; however, it significantly improved the ductility of the porous Ti64. Moreover, heat treatment improved the energy absorption efficiency of the porous Ti64. The samples heat-treated at 750 °C had the highest energy absorption of 233.6 ± 1.5 MJ/m^3^ at *ε* = 50%.

## 1. Introduction

Compared to fully dense materials, porous metallic materials have many excellent properties [1], such as low density, large specific surface area, good energy absorption, and excellent permeability; therefore, they are widely used in the aerospace [2], automobile [3], architecture, medical [4,5,6,7], environmental protection [8], and acoustics [9] industries. Moreover, porous structures are ubiquitous and have been found in bones, bamboo, wood, and coral. Unfortunately, the traditional subtractive manufacturing process is difficult to use for fabricating porous metallic structures with complex topographies. With the development of additive manufacturing, it has become possible to manufacture these complex porous structures precisely, thereby expanding the application area of these porous metallic materials. 

Selective laser melting (SLM) is a prevalent additive manufacturing process that fabricates porous titanium alloy structures because of its high material reuse efficiency, high flexibility, and precise shape production [10]. Furthermore, the titanium alloy Ti-6Al-4V has high specific strength, low density, high fracture toughness, and excellent corrosion resistance [11], which adds to its wide applicability in the manufacturing of porous structures. In recent years, there have been many studies [12,13,14,15] about porous Ti64 structures; most of the works are about the effects of topological morphology (different porosity, different pore size, or different pore unit) on the mechanical properties, permeability, and biological properties of porous structure. However, research on the effects of heat treatment processing on the microstructural and mechanical properties of additively manufactured porous Ti64 structure is still limited. On the other hand, it has already been confirmed that the heat treatment would improve the mechanical properties of SLM-built Ti64 parts by removing internal defects and residual stresses [16,17,18,19]. Moreover, the mechanical behavior of porous structures is not only related to its complex topology, but is also related to the microstructure of the material. It is crucial to consider both structural and microstructure of a material, while manufacturing porous structures for the specific application. 

The present work aims to study the effect of different heat treatment temperatures on the microstructural and mechanical properties of porous Ti64 structure. In this study, SLM was used to manufacture porous Ti64 structures. Optical microscope (OM) and X-ray diffraction (XRD) were used to determine the microstructures under different heat treatment conditions. Finally, a universal material testing machine was utilized to test the mechanical properties of the porous Ti64 structure.

## 2. Materials and Methods 

### 2.1. Design of Porous Structure

Figure 1 shows the design process of porous structures. The unit cell of the body-centered cubic (BCC) was modeled by Solidworks 2016 (Dassault Systemes, Walkerson, French). The unit cell size of BCC was 2 × 2 × 2 mm. In this work, four different porosities (46.3%, 56.3%, 66.2%, and 75.4%) of BCC were acquired by changing the diameter of the strut. The size of each sample was designed to be 6 × 6 × 12 mm. 

### 2.2. Materials and Manufacturing

The porous structures were fabricated by a selective laser melting machine S200 (Bright Laser technologies, Xi’an, China). The particle morphology of Ti64 was investigated by a scanning electron microscopy (SEM, PhenomXL, Shanghai, China). The Ti64 powder was nearly spherical with a smooth surface, as shown in Figure 2a. The particle size distributions ere d_0.1_ = 25.1 μm, d_0.5_ = 37.8 μm, and d_0.9_ = 56.5 μm, respectively. The Ti64 powder contained low levels of oxygen, nitrogen, carbon, and iron, and the chemical composition was as listed in Table 1, which is not much different from previous research [12]. During manufacturing, the SLM laser was rotated by 67° between two adjacent layers, the laser power was 100 W, and the layer thickness was 30 μm. The scanning strategy and manufacturing setup is shown in Figure 2b. 

The as-built samples underwent heat treatment at 750, 850, and 950 °C, respectively, at a heating rate of 10 °C/min; were held for 2 h; and were finally cooled to room temperature in a furnace. In order to prevent oxidation during heat treatment, the samples were sealed in vacuum tubes.

### 2.3. Microstructural Observation

In this study, the OM (Leica, DMI3000, Wetzlar, Germany) and SEM were used to observe the microstructures of the as-built and heat-treated samples. During observation, the surfaces of all samples were ground by SiC paper (600, 1500, and 2000) and mechanically polished, before being etched in Kroll solution (92 mL H_2_O, 6 mL HNO_3_ and 2 mL HF). XRD (Rigaku, Ultima, Japan) with a copper anticathode, 40 KV acceleration voltage, and 40 mA electric current, was applied in order to determine the phase composition of porous specimens.

### 2.4. Mechanical Testing

According to the international standard ISO 13314 for the compression test of porous metallic materials, the electronic universal testing machine (Shenzhen Kailiqiang Electronic Stretching Equipment Co., Ltd, WDW-300HC, Shenzhen, China) was used to determine the mechanical properties. The compression rate was set at 1 mm/min. Three samples of each group were tested; the experiment was terminated only when the sample was broken entirely or completely dense.

## 3. Results and Discussion

### 3.1. Morphology of Porous Ti64 Structure 

The as-built samples have a similar appearance in comparison to the design models, as shown in Figure 3. However, Figure 4 shows a very rough surface on the as-built strut; there are many unmolten or semi-melt spherical particles adhering to the strut surface; and the size of these particles is very close to that of the Ti64 powder in Figure 2a. Fortunately, the bound metal particles can be removed by sand blasting and post treatment [20].

The dry weight method was applied to determine the actual porosities of the as-built samples using the following equation:(1)$$P=\left(1-\frac{W_{p}}{W_{d}}\right)\times 100\%,$$
where *P* is the actual porosity (%); *W_p_* is the weight of as-built samples (g); *W_d_* is the equivalent mass (g); *W_d_* = ρ*V*; ρ is the density of Ti64, which is 4.51 g/cm^3^ [13]; and *V* is the measured volume of porous structure (cm^3^). The measured porosities and strut diameters of the as-built samples are shown in Table 2. It can be observed that there is still a particular deviation between the as-built samples and the theoretically designed models. The observed differences in porosity and strut diameter between each model and manufactured sample did not exceed 6%. The porosities measured by the dry weight method and designed model were very close. The actually manufactured porous samples had thicker struts than the models.

### 3.2. Microstructure

Figure 5 shows the microstructures of the porous Ti64 samples before and after heat treatment. The microstructure of the as-built sample is mainly composed of acicular α′martensite, because of the extremely high cooling rate during laser melting process, making the β domain phase transform to fine α′martensite [21]. Some pores were found in the as-built sample, as shown in Figure 5a,b. This defect may have been caused by the heat wave airflow that could not escape during the solidification, and this defect may have an unfavorable effect on the mechanical properties of as-built samples [22]. For the samples heat-treated at 750 °C and 850 °C, it is clear that the fine α′martensite began transforming to an α + β lamellar mixture microstructure, and α phases began to present a lath pattern, but there is still some needle-like α phase in their microstructures, as illustrated in Figure 5c–f. And comparing Figure 5 d–f, the microstructure is more uniform and the α lath is coarser for the samples heat-treated at 850 °C. In the case of the specimens heat-treated at 950 °C, the fine α′martensite completely transformed to a mixture of α + β, and the α phase showed a more obvious lath pattern, as shown in Figure 5g,h. In short, with the increase of the heat treatment temperature, the average width of the lamellar α phase gradually increased, as illustrated in Figure 6. When heat treatment temperature increased to 950 °C, the average width of α lath increased to 3.1 μm, as shown in Figure 6.

Figure 7 shows the XRD patterns of the as-built and heat-treated samples. According to pervious research [23], it is difficult to distinguish between α and α′phases, because those two phases have a similar hexagonal (hcp) structure; therefore, all peaks of as-built samples can also be considered α or α′phase. The XRD peak intensity of the β phase is relatively low for the samples heat-treated at 750 °C and 850 °C; it is even hard to see the presence of the β-peak. This indicates that these samples contain very low transformed β phase. And the β phase peak intensity is higher for the samples heat-treated at 950 °C. The content of β phase in this heat treatment condition is still low, according to previous research [17].

### 3.3. Mechanical Properties

#### 3.3.1. Compression Tests of Porous Structures

Figure 8 represents the stress–strain curves of the porous Ti64 structures with different porosities after heat treatment. According to Gibson and Ashby [1], it can be seen from the stress–strain curves that the deformation of the porous sample changed from brittle to ductile after heat treatment. A jagged plateau and fragmentation appear in the stress–strain curves of the as-built samples, meaning the as-built samples are brittle. The stress–strain curve of the heat-treated sample is relatively smooth, indicating that the strut of a heat-treated sample does not undergo severe fracture in the compression tests. After heat treatment, the acicular α′ martensite in the as-built samples transforms into a coarse α + β lamellar microstructure. The α + β lamellar microstructure can effectively prevent the crack’s growth under compressive conditions. In addition, the fracture morphology of the samples tested after compression in Figure 9 shows that all samples underwent the final compaction stage with a fracture zone of about 45° from the loading direction.

In this study, a stress–strain of 10% for the sample being tested was selected as the compressive strength. Table 3 compares the mechanical properties of porous Ti64 structures before and after heat treatment. It can be seen from Table 3 and Figure 10a that the compressive strength, platform stress, elastic modulus, and yield stress of the porous structure gradually decreased, when the porosity of the sample increased. At the same time, the compressive strengths and yield strengths of samples with the same porosity decreased as the heat treatment temperature increased. Among them, the samples with lower porosity had a more considerable reduction in yield strength, with the reduction of 19.9%, and the yield strength of samples heat treated by 950 °C decreased from 255.8 MPa to 204.9 MPa. However, from Figure 10b, it can be seen that the compression elongation of the sample increased with the increase of heat treatment temperature. The samples with lower porosity had a lower compression elongation. The maximum increase in compression elongation reached 14.7%, because the α′ phase in the as-built sample gradually transformed to coarse α + β lamellar microstructure. The lamellar microstructures can change the crack path of the sample during fracture and increase the compressive elongation of the heat-treated samples [18,24]. Under the same porosity, the heat treatment has little effect on the elastic modulus of samples with the same porosity, and the change in the elastic modulus of the samples after heat treatment was less than 8%.

#### 3.3.2. Gibson–Ashby Model

In previous studies [12,14,15,20], the Gibson–Ashby model has been extensively used to predict the relationship between the porosity of a porous structure and its elastic modulus and yield strength. The relationship can be expressed by the following formula:(2)$$\frac{E_{p}}{E_{s}}=C_{1}{\left(1-P\right)}^{a}$$
(3)$$\frac{\sigma_{p}}{\sigma_{s}}=C_{2}{\left(1-P\right)}^{b},$$
where *E_s_* and *σ_s_* are the elastic modulus (GPa) and yield strength (MPa) of the Ti64; *E_p_* and *σ_p_* are the elastic modulus (GPa) and yield strength (MPa) of the porous structure, respectively. *P* is the porosity of the porous structure (%); *C*_1_, *C*_2_ are the geometric proportionality constants of the porous structure. According to previous studies [12,25], the exponent factors a and b of porous structure are related to their topological morphology. In this study, the density of Ti64 is 4.51 g/cm^3^ [13], the elastic modulus is 110 GPa [26], and the yield strength is 1096 MPa [11].

According to Gibson et al. [1], the values of these four constants were *C*_1_ = 1.0, *C*_2_ = 0.3, *a* = 2.0, and *b* = 1.5. In our work, the predicted values were higher than the above values, as shown in Figure 11. As shown above, heat treatment has little effect on the elastic modulus of a porous Ti64 structure with the same porosity. According to the calculations, *C*_1_ = 0.1888 and *a* = 2.0 in this study. The exponent factor of a was the same as that of Crupi et al. [12] but their *C*_1_ magnitude was 0.1453. It can be seen that the relationship between porosity and the elastic modulus of the BCC porous Ti64 model predicted by this study is consistent with the research of Crupi et al. [12]; the predicted result is shown in Figure 11a. 

While the heat treatment temperature increased, the yield strength of the porous structure with the same porosity gradually decreased. Therefore, in this study, four prediction models for the relationship between yield strength and porosity were obtained, and the predicted curves of Crupi et al. [12] were closely to this work, as shown in Figure 11b. It can be observed that as the heat treatment temperature increased, the *b* value in the yield strength prediction model increased from 2.0 to 2.5, and the curve of the yield strength prediction model gradually moved down.

#### 3.3.3. Energy Absorption Characteristics

According to the international standard ISO 13314, the formulas for the energy absorption and energy absorption efficiency of porous structures re as follows:(4)$$W=\frac{1}{100}\int_{0}^{\varepsilon }\sigma d\varepsilon$$
(5)$$W_{e}=\frac{W}{\sigma_{max}\times \varepsilon }\times 10^{4},$$
where *W* is the energy absorption by the porous structure (MJ/m^3^), *W_e_* is the energy absorption efficiency of the porous structure (%), *ε* is the compressive strain (%), and is *σ* the maximum compressive strength (MPa) of the porous structure within a limited strain range. 

The energy absorption capabilities of the porous Ti64 structures at different heat treatments are shown in Figure 12. It was observed that the samples with the same porosity heat-treated at 750 °C had better energy absorption capability. With the increase of heat treatment temperature, the energy absorption ability of the samples gradually decreased. Particularly, the samples with porosity of 46.3% had the highest energy absorption at *ε* = 50% (233.6 ± 1.5 MJ/m^3^) after heat-treated at 750 °C. In this study, a strain of 50% was selected to calculate the specific energy absorption of porous structures; and the samples with low porosity have higher energy absorption energy, as shown in Figure 12e. It is because the α′martensite in as-built samples transforms into a lamellar α + β mixed microstructure, these microstructures can make the cracks expand differently along the α lath during the deformation process of the sample, so that the energy absorption capacity of the sample is improved. And increase of the heat treatment temperature makes the width of α lath larger, which reduces the compressive strength and decreases the energy absorption capacity of the porous structure. Beyond that, due to the higher compressive strength of the sample with low porosity, the energy absorption capacity of samples with low porosity is greater. It is believed that the specific heat treatment can improve energy absorption capability of porous Ti64 structures.

Figure 13 shows the energy absorption efficiency curve of the porous Ti64 structures with different porosities after different heat treatments at compressive strain between 5% and 50%. It was found that the energy absorption efficiency curve of porous Ti64 structures has a good correspondence with their stress–strain curve. All these samples have a rising stage, stable stage, and dropping stage. Meanwhile, for samples with porosities of 46.3% and 56.3%, they have second rising stage; this is due mainly to these samples breaking secondarily during densification. The maximum compressive strength remains the same, but the strain still increases before compressive strain 50%, as shown in Figure 8. In the stable stage, the porous Ti64 structures have the highest energy absorption efficiency, but the efficiency decreases quickly after that. The curve of the as-built sample curve has jagged bumps during the stable stage, and as the heat treatment temperature increases, the curve tends to stabilize. This was because that the as-built samples were brittle and because their strut was broken in the stable stage. After heat treatment, the samples were more ductile; the struts were not easy to break. It is noteworthy that the samples with porosities 46.3%, 56.3%, and 66.2% had maximum efficiency in the stable stage; the heat treatment would slightly decrease their maximum efficiency. However, before the strain 15%, the heat-treated samples’ efficiencies were higher than those of the as-built samples. And when the strain exceeded 15%, the curve fluctuated to some extent, which means the porous structure strut begin began to break or crack. It can be seen that the heat treatment can improve the energy absorption efficiency of a porous Ti64 structure before it is broken. Meanwhile, for the porous Ti64 structures with low porosity, when their struts begin to crack, efficiency of the as-built samples would higher than that of the heat-treated samples.

## 4. Conclusions

The present work investigated the effect of heat treatment temperature on the microstructural and mechanical properties of porous Ti64 structures. The main conclusions are the following:(1)The microstructure of the as-built samples is mainly composed of needle-shaped α′ martensite. After heat treatment at 750 °C, the needle-shaped α′ martensite transformed into lamellar α + β microstructure. As the heat treatment temperature increased from room temperature to 950 °C, the width of lamellar α phase gradually increased from 0.3 to 3.1 μm.(2)The increase in heat treatment temperature could gradually reduce the compressive strength and yield strength of the porous Ti64 structure. The mechanical properties of the porous structures with low porosity were more sensitive to the heat-treatment temperature, but the heat treatment processing hardly affected the elastic moduli of the porous structures.(3)The energy absorption condition of porous Ti64 structure could be more stable after heat treatment. Heat treatment at 750 °C improved the energy absorption capabilities of the porous Ti64 structures. The samples with porosity of 46.3% had the highest energy absorption at *ε* = 50% (233.6 ± 1.5 MJ/m^3^) after being heat-treated at 750 °C for 2 h.

## Figures and Tables

**Figure 1 materials-13-00593-f001:**
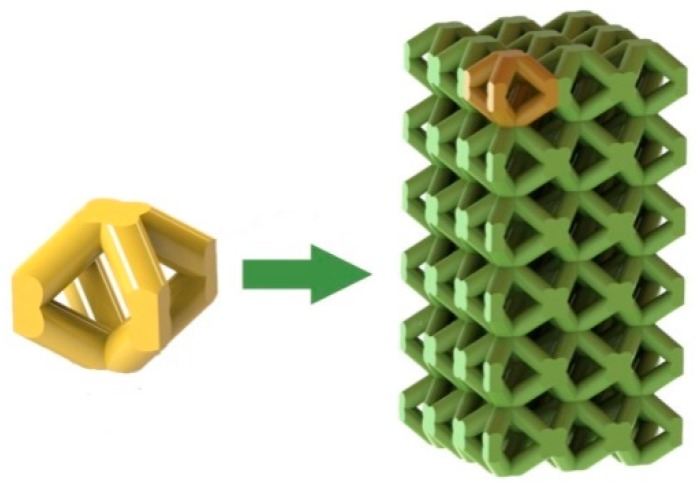
Schematic of designing process for the body-centered cubic (BCC) porous structure.

**Figure 2 materials-13-00593-f002:**
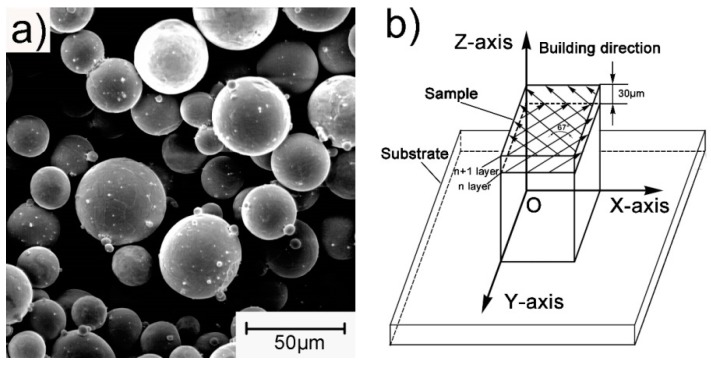
(**a**) SEM image of the spherical Ti-6Al-4V powder for selective laser melting (SLM) manufacturing; (**b**) schematic of the SLM manufacturing process.

**Figure 3 materials-13-00593-f003:**
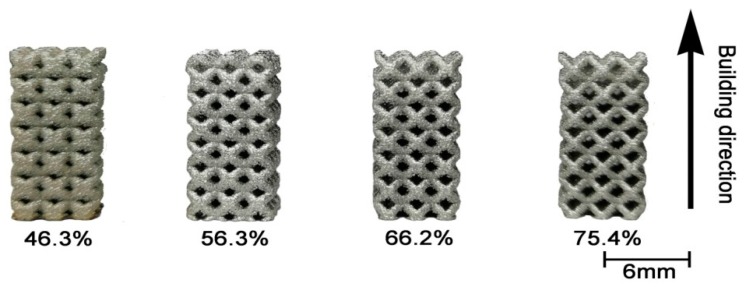
The as-built Ti64 samples with different porosity designs.

**Figure 4 materials-13-00593-f004:**
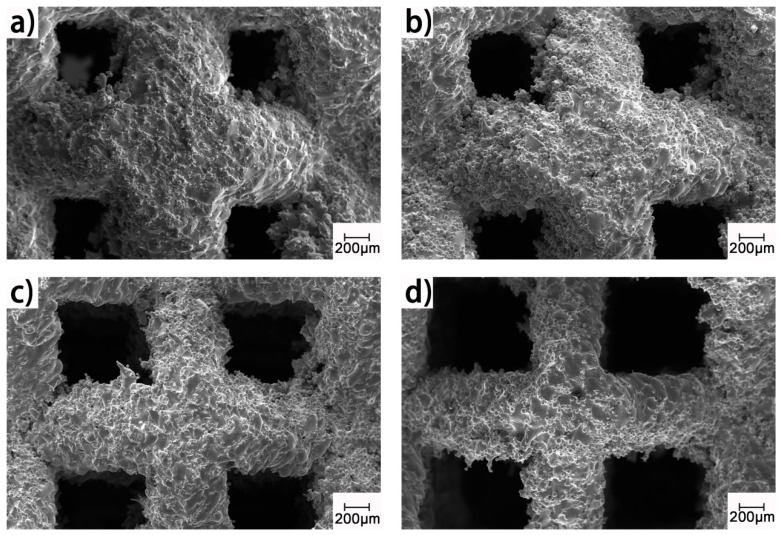
SEM images of the as-built porous samples with different strut diameters: (**a**) 780 μm; (**b**) 680 μm; (**c**) 580 μm; (**d**) 480 μm.

**Figure 5 materials-13-00593-f005:**
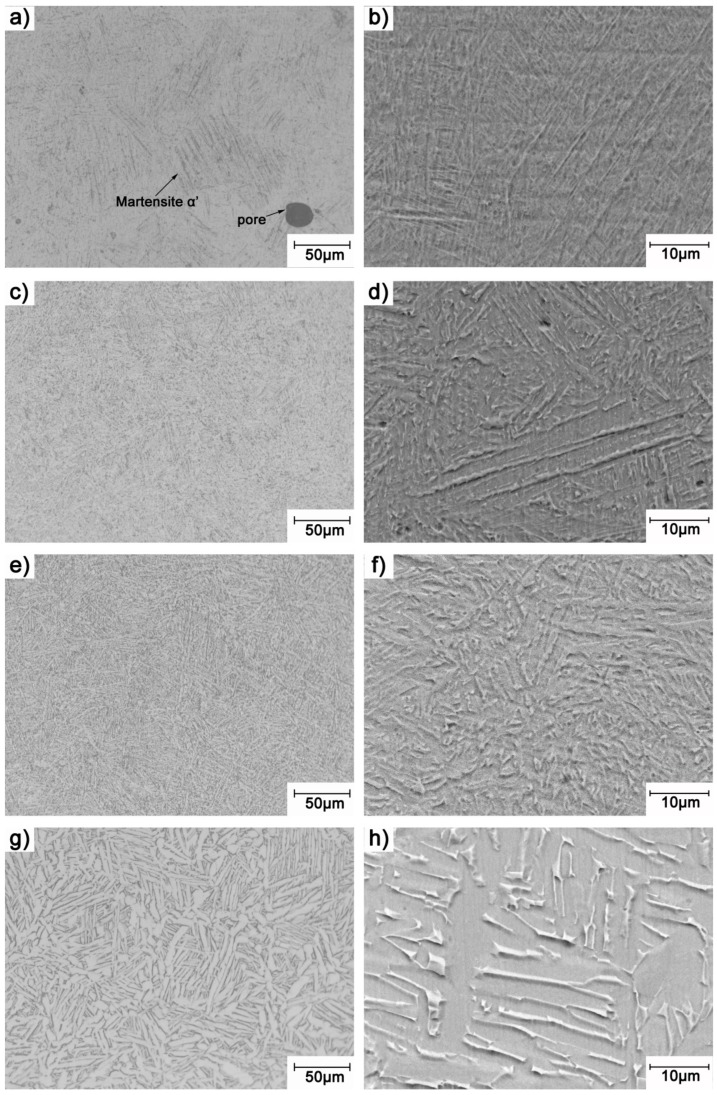
Microstructure of porous Ti64 before and after heat treatment: (**a**) as-built OM image; (**b**) as-built SEM image; (**c**) heat-treated at 750 °C OM image; (**d**) heat-treated at 750 °C SEM image; (**e**) heat-treated at 850 °C OM image; (**f**) heat-treated at 850 °C SEM image; (**g**) heat-treated at 950 °C OM image; (**h**) heat-treated at 950 °C SEM image.

**Figure 6 materials-13-00593-f006:**
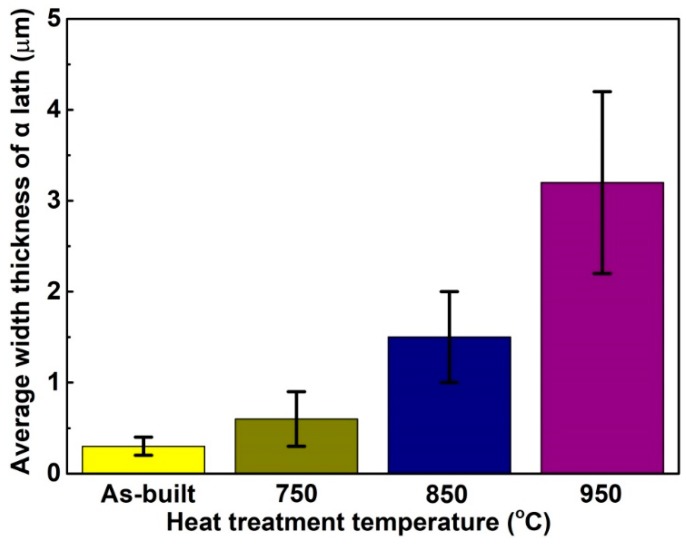
Average width of the α lath of porous Ti64 after different heat treatments.

**Figure 7 materials-13-00593-f007:**
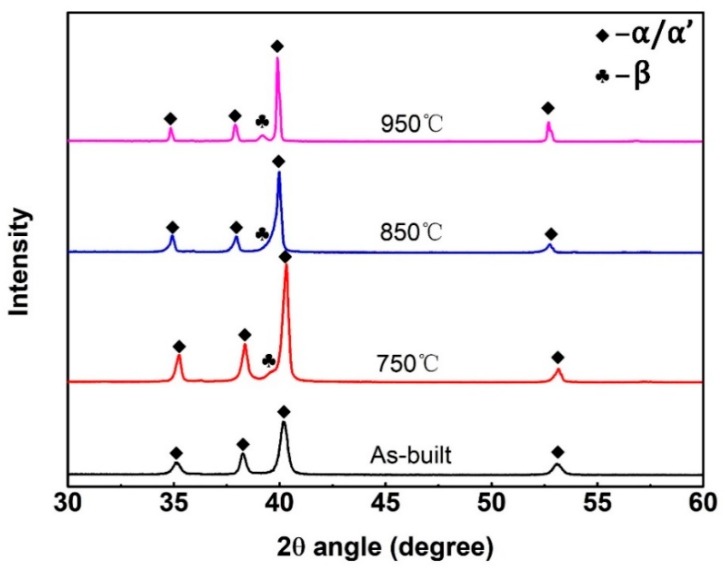
XRD pattern of the porous Ti64 heat treated under different temperatures.

**Figure 8 materials-13-00593-f008:**
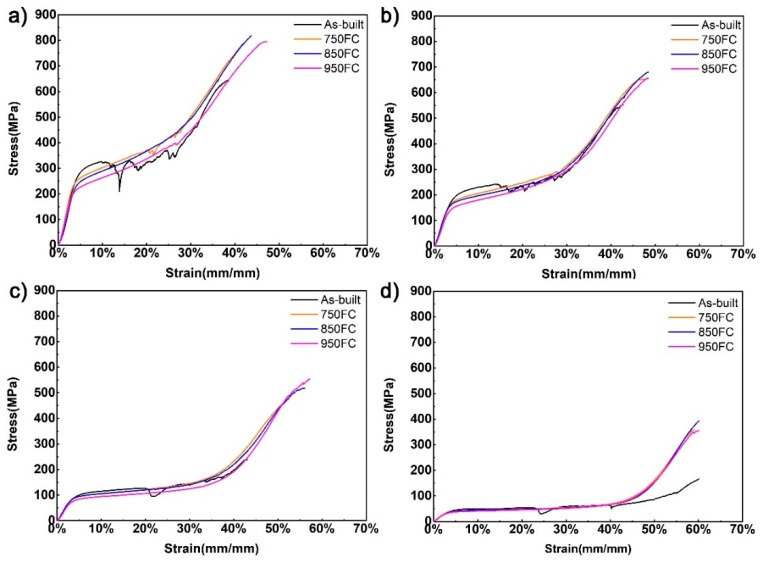
Compression stress–strain curves of porous Ti64 structures with different porosities after different heat treatments: (**a**) 46.3%; (**b**) 56.3%; (**c**) 66.2%; (**d**) 75.4%.

**Figure 9 materials-13-00593-f009:**
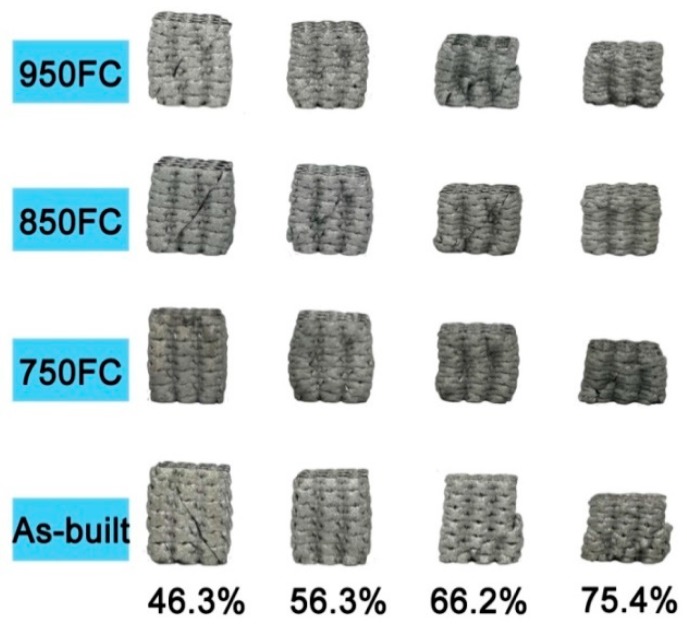
Fracture morphology of the compressive samples after compression.

**Figure 10 materials-13-00593-f010:**
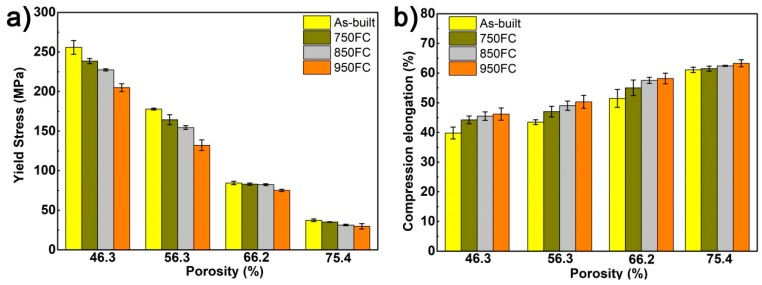
Compressive yield strengths (**a**) and compressive elongations (**b**) of the porous Ti64 samples under different heat treatments.

**Figure 11 materials-13-00593-f011:**
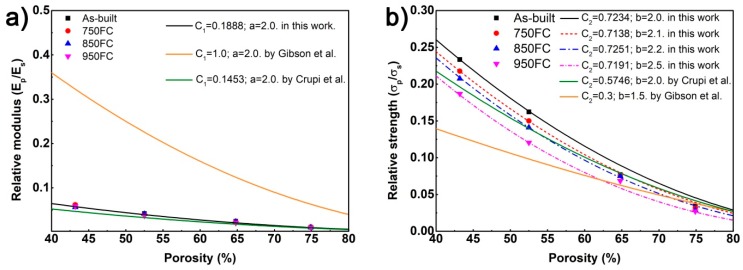
Relationship between the porosity of porous structure (**a**) and its elastic modulus and compressive strength (**b**).

**Figure 12 materials-13-00593-f012:**
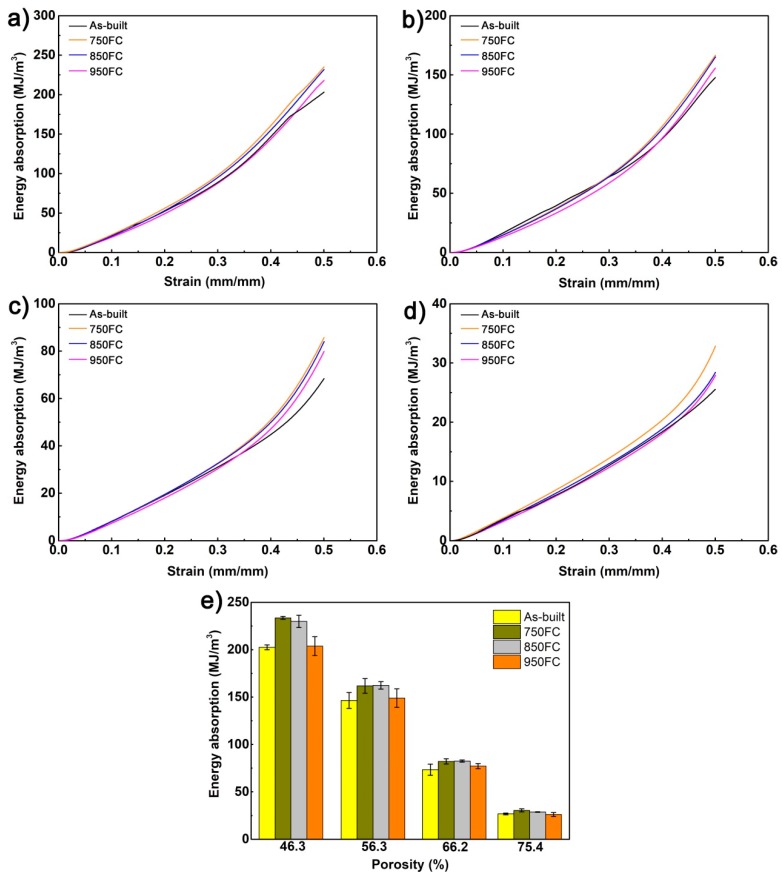
The energy absorptions of porous Ti64 structures with different porosities before and after different heat treatments: (**a**) 46.3%; (**b**) 56.3%; (**c**) 66.2%; (**d**) 75.4%; (**e**) energy absorption at *ε* = 50%.

**Figure 13 materials-13-00593-f013:**
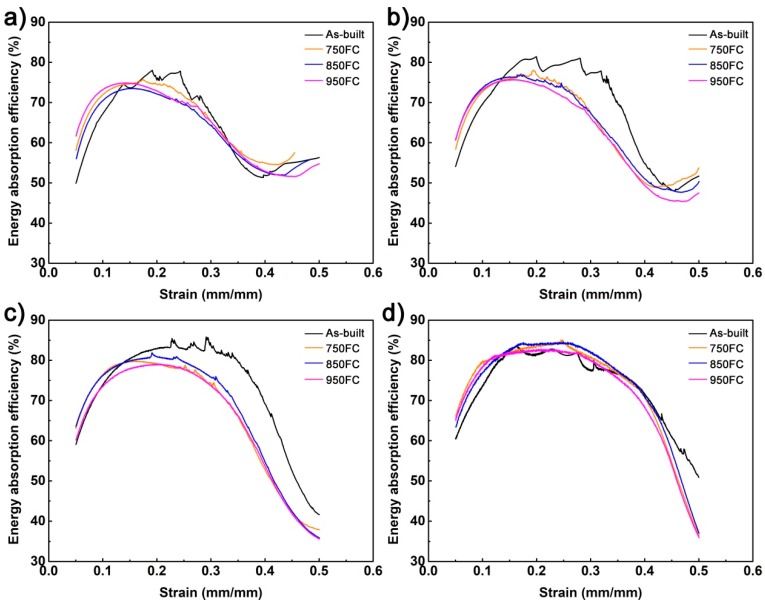
The energy absorption efficiency curves of porous Ti64 structures with different porosities after different heat treatments: (**a**) 46.3%; (**b**) 56.3%; (**c**) 66.2%; (**d**) 75.4%.

**Table 1 materials-13-00593-t001:** Chemical compositions of Ti-6Al-4V powder.

Element	Ti	Al	V	Fe	C	H	O	N
Mass (%)	Bal	5.5	3.5	<0.3	<0.08	<0.015	<0.2	<0.05

**Table 2 materials-13-00593-t002:** Strut diameters and porosities of designed and manufactured porous Ti64.

Sample	Design Strut Diameters (μm)	Actual Strut Diameter (μm)	Design Porosity (%)	Actual Porosity (%)
1	780	802 ± 5	46.3	43.2 ± 0.5
2	680	721 ± 2	56.3	52.5 ± 1.5
3	580	611 ± 3	66.2	64.8 ± 0.5
4	480	508 ± 3	75.4	74.9 ± 0.1

**Table 3 materials-13-00593-t003:** Mechanical properties of porous Ti64 structure after different heat treatments.

Sample	Heat Treatment Stemperature (°C)	Design Porosity (%)	Elastic Modulus (GPa)	Compression Stress (MPa) *ε* = 10%	Plateau Stress (MPa) *ε* = 20–30%
1	0	46.3	6.7 ± 0.7	320.6 ± 9.4	365.3 ± 12.3
2		56.3	4.6 ± 0.1	228.2 ± 2.4	252.1 ± 7.6
3		66.2	2.7 ± 0.4	112.9 ± 4.6	126.1 ± 7.4
4		75.4	1.2 ± 0.1	47.3 ± 3.1	48.6 ± 1.5
5	750	46.3	6.8 ± 0.4	300.6 ± 3.6	420.4 ± 2.6
6		56.3	4.4 ± 0.1	202.5 ± 0.1	265.2 ± 15.9
7		66.2	2.6 ± 0.1	104.4 ± 1.7	128.9 ± 4.1
8		75.4	1.3 ± 0.2	44.7 ± 0.9	52.0 ± 1.2
9	850	46.3	6.2 ± 0.2	287.8 ± 2.7	422.1 ± 2.3
10		56.3	4.5 ± 0.3	196.2 ± 3.5	263.4 ± 10.1
11		66.2	2.6 ± 0.1	106.0 ± 0.4	130.0 ± 1.1
12		75.4	1.2 ± 1.1	41.6 ± 4.0	46.3 ± 6.4
13	950	46.3	6.4 ± 0.7	265.9 ± 0.3	395.3 ± 22.5
14		56.3	4.1 ± 0.4	173.4 ± 13.0	241.7 ± 26.0
15		66.2	2.3 ± 0.1	96.6 ± 2.5	119.9 ± 4.5
16		75.4	1.1 ± 0.1	36.4 ± 6.6	45.8 ± 3.2

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
