# Peer review of "The Effects of Post Heat Treatment on the Microstructural and Mechanical Properties of an Additive-Manufactured Porous Titanium Alloy"

_materials, 2020, doi:10.3390/ma13030593_

Round 1

Reviewer 1 Report

The paper focuses on the relevant issue that is investigation of post heat treatment effect on microstructure and mechanical properties of additive manufactured porous titanium alloy.

The presented results are of scientific and practical value. 

However, there are a number of comments to the paper. Please consider as given below:

Add the measurement error to Table 2.

Increase the quality of Figure 5.

Add comments to Figures 5 a-d.

It is necessary to add explanations (with references to the literature) on the change in the microstructure during heat treatment, on the influence of heat treatment and porosity on the mechanical properties of Ti64.

Reviewer 2 Report

A general comment. Selective laser melting is a trade-marked name, which should only be used for materials manufactured by machines from SLM-solutions. I agree that SLM is widely used within the community to talk about laser-based powder bed fusion (I've used it extensively as well). But to stick to the rather new standard, LPBF (laser powder bed fusion) is more correct.  

The chapter result should be called result and discussion, 'cause both are mixed. 

Row 42: Additively not additive

Row 49: Temperatures

Row 50: My personal opinion is not to use "we", instead use "were used" as in row 51. Ny native language isn't English though

Row 52-53: Finally, a universal material testing machine was utilized to test the mechanical properties of the porous Ti64 structure.

Row 55: title indent

Row 55-59: Calling the porous structure BCC can make it a bit confusing? One thinks about the beta phase when hearing BCC. 

Row 70: "are shown". Rephrase the sentence. The image doesn't show all the relevant process parameters, rather the setup

Table 1: What the purpose of this table? It doesn't show the chemical composition of the powder that was used

Row 81: an OM

Figure 5 a): Not very convinced that it's a pore? Could just be from the etching. If you wanna examplify pores I suppose there are better examples in your material? The magnification images can be increased in size. The needles in a) can't be seen that clearly. 

Row 129: I've been taught to use shown instead of seen, like you do at row 125, yet again english isn't my native language

Row 131: Cite a reference for 995C

Row 132: "It was shown that α' martensite started to decompose into α+β structure, when heat treatment was above 650 ℃ [22]." A bit confusing? You mix your result with a reference?

You have a novel SLM (LPBF) machine, that I haven't heard about before. Would be interesting if you just to compare your alpha needle measurements with previous published results for as-built, very briefly. I have as an example done similiar measuements like you, see: https://link.springer.com/article/10.1007/s00170-019-04002-8

Row 171: which can reduce slip resistance, how?

Row 231: remove "shown" or "illustrated"

Row 232: , should be .

Row 240: The

Overall a good contribution. Well written and of interest for porous structures. Good job!
